# Nitrous Oxide Emission from Forage Plantain and Perennial Ryegrass Swards Is Affected by Belowground Resource Allocation Dynamics

**John Kormla Nyameasem** [1,2,*], **Enis Ben Halima** [1], **Carsten Stefan Malisch** [1], **Bahar S. Razavi** [3], **Friedhelm Taube** [1,4] and **Thorsten Reinsch** [1]

[1] Grass and Forage Science/Organic Agriculture, Christian-Albrechts University, 24118 Kiel, Germany; stu112657@mail.uni-kiel.de (E.B.H.); cmalisch@gfo.uni-kiel.de (C.S.M.); ftaube@gfo.uni-kiel.de (F.T.); treinsch@gfo.uni-kiel.de (T.R.)

[2] Council for Scientific and Industrial Research (CSIR), Animal Research Institute, Achimota, P.O. Box AH20, Ghana

[3] Department of Soil and Plant Microbiome, Institute of Phytopathology, University of Kiel, 24118 Kiel, Germany; brazavi@phytomed.uni-kiel.de

[4] Grass Based Dairy Systems, Animal Production Systems Group, Wageningen University (WUR), 6708 PB Wageningen, The Netherlands

\* Correspondence: jnyameasem@gfo.uni-kiel.de

**Abstract:** Soil–plant interactions affecting nitrous oxide ($N_2O$) are not well-understood, and experimental data are scarce. Therefore, a greenhouse experiment was conducted in a $3 \times 3$ full factorial design, comprising three mineral N fertilizer rates (0, 150 and 300 kg N ha$^{-1}$) applied to monoculture swards and a binary mixture of *Plantago lanceolata* and *Lolium perenne*. The parameters measured included daily $N_2O$ emissions, aboveground (AG) and belowground biomass (BG), N and C yields, as well as leucine aminopeptidase (LAP) activity in the soil as an indicator for soil microbial activity. Nitrous oxide emission and LAP were measured using the static chamber method and fluorimetric microplate assays, respectively. Cumulative $N_2O$ emissions were about two times higher for *P. lanceolata* than *L. perenne* monoculture swards or the mixture ($p < 0.05$). The binary mixtures also showed the highest N use efficiency and LAP activity, which significantly ($p < 0.05$) correlated with the C concentration in the belowground biomass. *Plantago lanceolata* was generally ineffective at reducing $N_2O$ emissions, probably due to the young age of the swards. Among the biological factors, $N_2O$ emission was significantly associated with biomass productivity, belowground C yield, belowground N use efficiency and soil microbial activity. Thus, the results suggested belowground resource allocation dynamics as a possible means by which swards impacted $N_2O$ emission from the soils. However, a high N deposition might reduce the $N_2O$ mitigation potential of grasslands.

**Keywords:** fertilization; nitrous oxide; perennial ryegrass; forage plantain; nitrification inhibition; Northern Germany

## 1. Introduction

Grasslands provide various ecosystem services, such as carbon (C) sequestration, soil erosion prevention, landscape and biodiversity protection. However, they are a significant source of anthropogenic greenhouse gases, including nitrous oxide ($N_2O$), especially when utilized for ruminant livestock production [1]. Nitrous oxide is a greenhouse gas (GHG) with a global warming potential (GWP) of 265–298 [2] and stratospheric ozone depletion potential [3], with agriculture contributing about 60% to global anthropogenic $N_2O$ emissions [4]. Nitrous oxide is produced through nitrification and denitrification processes facilitated by soil microbes. Thus, nitrous oxide emission is affected by heterogeneous factors, including climatic factors, soil properties, the soil microbiome and botanical factors [5–7], making the flux estimates highly uncertain [1]. Moreover, the functional role of

pastures in the soil–plant interactive processes leading to $N_2O$ production and emission is often omitted from predictive models. This is because the mechanisms by which plants reduce $N_2O$ emissions are not well-understood [8], resulting from insufficient experimental data.

The optimization of agroecosystems to retain more N under intensive management has generally focused on aboveground traits and less on belowground plant–soil interactions [9]. Meanwhile, plant roots play critical roles in ecosystem processes, including nutrient cycling and $N_2O$ production [10], and such functions could be more relevant than leaf traits in explaining N cycling processes [11]. Some botanical factors that affect the $N_2O$ mitigation potential of pastures may include belowground species diversity [12], the presence or absence of species capable of nitrification inhibition (NI) [11], rhizosphere priming effects and whether the species are conservative or acquisitive [9]. Nitrification-inhibiting species in mixed swards are particularly appealing and have received attention, as they can concomitantly provide high-quality forage and reduce negative environmental effects.

One of the most promising species in previous studies has been ribwort plantain (*Plantago lanceolata*), which has exhibited the potential to reduce $N_2O$ emission in grazed pastures [13–15]. However, the results have not been consistent, probably due to the numerous factors that affect $N_2O$ emission. Nevertheless, de Klein et al. [8] suggested that a key mechanism by which plantain affects the $N_2O$ emission factor could be root exudates that inhibit nitrification and/or increase N immobilization. A previous study [9] identified high biomass production and fast N uptake as the main pathways by which plants impact $N_2O$ emission, and they mentioned specific leaf area and root length density as particularly relevant determinants. However, the previous authors only based their conclusions on swards involving grasses without known high NI potential species.

In any case, differences in resource allocation belowground by pastures could impact the soil microbial activity, with important implications for nutrient cycling and ecosystem functioning [16]. Soil microbiomes are sensitive to changes in their environment and may express enzymes in response to soil N availability. These extracellular enzymes, such as leucine aminopeptidase (LAP)—a hydrolytic enzyme involved in N cycling and splitting of N containing polymeric compounds—are produced to catalyze chemical reactions that facilitate soil organic matter degradation to acquire the necessary resources for microbial metabolic function [17]. The expression of such N acquiring enzymes varies according to N concentrations in the soil [17]. While the factors that affect microbes' role in $N_2O$ emission are numerous and complex, their population dynamics and expression of enzymes could be essential biomarkers for $N_2O$ emission [18]. A larger microbial population encompasses a greater nitrifying and denitrifying activity, but their effect on $N_2O$ emissions might depend on the prevailing environmental conditions. For instance, an increased soil N supply decreased the soil pH, C availability and water content and generally increased the $N_2O:N_2$ ratio under anaerobic soil conditions [19]. On the other hand, a higher C supply to soil microbes could reduce the available N through immobilization, particularly in aerated soils [20]. A large microbial population in the soil might lead to an increased consumption or immobilization of surplus N in sandy soils, especially when N-acquiring microbes are involved, leading to reduced $N_2O$ emission [8].

The most sown grassland species in Europe is perennial ryegrass (*Lolium perenne*) [9] due to its persistence under frequent harvesting and the production of high-quality forage, and combining it with other species can enhance the productivity of both grassland and animals while also reducing the negative environmental impacts [9]. Mixed pastures have faster growth and rapid resource acquisition rates than monoculture pastures, increasing plant productivity through increased niche complementarity and resource uptake efficiency [12]. In addition, the presence of small herbs in the pasture might increase microbial $NH_4^+$ consumption and gross inorganic N immobilization due to increased rhizodeposition and microbial growth stimulation [21]. Accordingly, mixed pastures have a higher tendency to mitigate $N_2O$ emissions [7,12]. Understanding the impact of plant functional traits on $N_2O$ emissions is vital to improving the estimates of $N_2O$ emissions from pastures [8]. Ac-

cordingly, a greenhouse pot experiment was designed with a constant soil temperature and moisture and three different mineral fertilizer rates to evaluate the effects of monoculture swards and a mixture of *L. perenne* and *P. lanceolata* on the $N_2O$ emission and aboveground (AG) and belowground (BG) C and N partitioning. The following hypotheses were tested using this setup: (i) the $N_2O$ emission from *P. lanceolata* swards is lower than from *L. perenne* monoculture swards due to their expected nitrification inhibiting effect, (ii) belowground biomass productivity and C yield significantly affect the $N_2O$ emission from soils under pastures and (iii) high soil microbial activity negatively affects $N_2O$ emission from soils and could, therefore, be a suitable biomarker for $N_2O$ emission from pastures.

## 2. Materials and Methods

### 2.1. Experimental Design

The experiment was conducted in a greenhouse located in the University of Kiel, Northern Germany (54°20′ N, 10°6′ E). The soil, obtained from grazed grass–clover swards from an organic experimental dairy farm, was a sandy loam and contained 2.0% organic matter, 0.14% N, C/N ratio of 8.4, 310-mg $P_2O_5$ kg$^{-1}$ dry soil, 120-mg $K_2O$ kg$^{-1}$ dry soil and a pH of 7.2 (Appendix A, Table A1). The soil was sieved through a 3-mm mesh, homogenized and placed in circular-shaped pots, measuring 0.14 m in diameter and 0.18 m in depth, and the bulk density uniformly set to 1.5 g/cm$^3$. This comparably small size allowed for a better root/soil ratio while being large enough to provide an adequate rooting volume for the duration of the experiment, even for the deep-rooting *P. lanceolata*. Before sowing, the soils were watered for about two weeks to activate them. The botanical composition was varied and included monoculture swards of *L. perenne* and *P. lanceolata*, and their binary mixture. *Plantago lanceolata* seeds were pregerminated in Petri dishes, and one viable plant was transplanted into each pot of *P. lanceolata* treatment. This method was adopted to ensure an even distribution of *P. lanceolata* in all the pots. The sowing density of *L. perenne* was the equivalent of 32 kg ha$^{-1}$ for all *L. perenne* containing pots. Before starting the gas measurements, all the plants were allowed to grow for approximately one month to ensure viable swards; after which, a standardization cut was performed.

The experimental units consisted of a 3 × 3 factorial design, comprising three N fertilizer (urea) rates (0, 150 and 300 kg N ha$^{-1}$ equivalents, subsequently called N0, N150 and N300) applied in three splits at intervals of about four and three weeks for the second and third applications, respectively (see Figure 1). The fertilizer was dissolved in 30 mL of water and applied by sprinkling from above. The three sward types (*L. perenne* (LP), *L. perenne* + *P. lanceolata* (LP + PL) and *P. lanceolata* (PL)) were replicated four times and arranged in a completely randomized block design. The greenhouse was illuminated for 16 h each day, with daytime temperatures set to 20 °C and nighttime temperatures to 15 °C. The soil moisture (volumetric) was maintained between 0.290 and 0.330 cm$^3$/cm$^3$ (i.e., 66.8–76.0 water-filled pore space) via capillary irrigation (Ortmann, Votloh, Germany) to maintain adequate soil moisture for optimal growing conditions and denitrification. It was monitored continuously using 5TM volumetric water content sensors coupled with EM 60 loggers (Meter Group, Pullman, WA, USA).

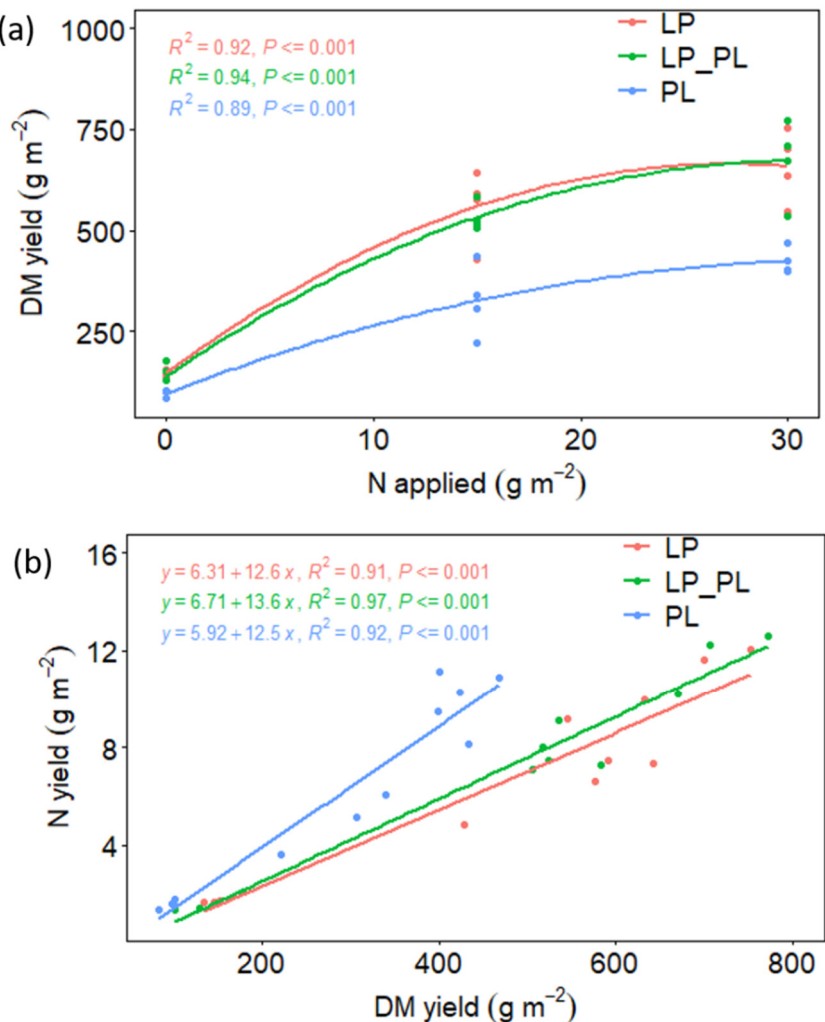

**Figure 1.** Graphs showing (**a**) the dry matter (DM) yield of pastures in response to varying N rates and (**b**) the linear relationships between the DM yield and nitrogen yield affected by the different swards.

### 2.2. Gas Sampling and Nitrous Oxide Measurement

Gas sampling started on the 5th of March 2020 after successfully establishing the swards and before the first fertilizer application and ended on the 27th of May 2020. After the first fertilizer application, the subsequent gas sampling was done on the 23rd of March and continued at intervals of 1–4 days, being more frequent immediately after each fertilizer application. Gas sampling was done using static chambers (height = 0.19 m) with the chamber collar sealed using a waterlogged basement and one air sample taken with a 30-mL polypropylene syringe from the chamber top at 0, 20, 40 and 60 min after closure. Each gas sample was transferred to a 12-mL pre-evacuated glass vial (Labco, High Wycombe, UK) and analyzed for $N_2O$ through a gas chromatograph (SCION 456-GC, Bruker, Leiderdorp, The Netherlands) equipped with a 63Ni electron-capture detector. Helium was used as the carrier gas and argon–methane as the make-up gas. Three certified $N_2O$ gas standards (300, 620 and 1510 ppb) were measured regularly to determine the analytical precision (standard deviation of 10 repeated measurements < 3 ppb). The gas samples were injected using an auto-sampler (model 271 LH, Gilson Inc., Middleton, WI, USA). The gas chromatograph procedure adopted a detector temperature of 320 °C, a column temperature of 40 °C and an injector temperature of 200 °C. Gas sampling was always carried out between 10:00 and 12:00 throughout the experiment. $N_2O$-N fluxes were calculated by linear regression between the measured $N_2O$-N concentrations and sampling time [22]. Accumulated $N_2O$ emissions over the experimental period (83 days) for each

treatment and replicate were calculated by linear interpolation between the consecutive measurements. The nitrous oxide emission factor (N$_2$ON EF) was calculated using the IPCC [23] recommended method (Equation (1)):

$$EF = \frac{N_2ON_{Treatment} - N_2ON_{Control}}{N \text{ applied}} \times 100\% \qquad (1)$$

### 2.3. Biomass Sampling and Chemical Analyses

The aboveground biomass (AG) was harvested at a fixed time of three weeks after the stabilization cuts. The plants were harvested by cutting AG from each pot at a 5-cm sward height. At the end of the pot experiment, the sward stubble was removed from the soil surface. The belowground biomass (BG) was separated from the soil with a hydropneumatic elutriation system [24] using a 0.63-mm sieve to separate the roots from the soil particles. Floating fine roots were collected with tweezers. All the plant samples (AG, stubble and BG) were oven-dried to a constant weight at 58 °C for 48 h to determine the DM yield. Each dried sample was milled through a 2-mm sieve (Cyclotech Mill, Foss Analytical, Hilleröd, Denmark), and then, the N and C contents were measured using a C/N analyzer (Vario Max CN, Elementar, Hanau, Germany). The ratio of N uptake by plants to fertilizer N input constituted N uptake efficiency (NUpE), and the N use efficiency (NUE) was calculated as (treatment N uptake—control)/N applied.

### 2.4. Soil Sampling and Measurement of LAP Activity

Soil samples were obtained at each biomass harvest using procedures described by Pijlman et al. [25] and Roseaux et al. [26]. Three soil cores (0–10-cm depth and, 1-cm diameter) were randomly sampled from each pot and mixed to obtain a composite sample. A subsample (~2 g) was then taken, immediately stored on dry ice and later transferred to a −20 °C freezer for the microbial enzyme analysis. The resulting hole was filled with a 10-mm diameter PVC pipe and closed off on top to prevent air and light interactions. The kinetics of hydrolytic enzymes involved in the N cycle were measured by fluorimetric microplate assays of 7-amino-4-methyl coumarin (AMC). The leucine aminopeptidase and tyrosine aminopeptidase activities were measured using L-leucine-7-amino-4-methyl coumarin (AMC-L) and L-tyrosine-7-amido-4-methyl-coumarin (AMC-T). All substrates and chemicals were purchased from Merck (Darmstadt, Germany). Microbial enzyme activities were determined in a range of substrate concentrations from low to high (0, 10, 20, 30, 40, 50, 100 and 200 μmol g$^{-1}$ soil). Saturation concentrations of the fluorogenic substrates were determined in preliminary experiments. Suspensions of 0.5-g soil (dry weight equivalent) with 50 mL of water were prepared using low-energy sonication (40-J s$^{-1}$ output energy) for 2 min. Then, 50 μL of soil suspension was added to 100 μL of substrate solution and 50 μL of buffer (TRIZMA (pH: 7.2) buffer for the AMC substrate) in a 96-well microplate. The fluorescence was measured in microplates at an excitation wavelength of 355 nm and an emission wavelength of 460 nm and slit width of 25 nm, with a Victor 3 1420-050 Multi-Label Counter (Perkin Elmer, Waltham, Massachusetts, United States of America). All enzymes were determined and incubated at the exact temperature within 2 h. After each fluorescence measurement (i.e., after 30 min, one h and two h), the microplates were promptly returned to the climate chambers so that the measurement time did not exceed 2–2.5 min. The assay of each enzyme at each substrate concentration was replicated three times in each plate. Additionally, each enzyme's assay at each substrate concentration was performed in three analytical replicates (12 wells in the microplate) for all four incubation replicates. The Michaelis–Menten equation (Equation (2)) was used to determine the parameters of the enzyme activity (*V*):

$$V = \frac{V_{max}\,[S]}{K_m + [S]} \qquad (2)$$

where $V_{max}$ is the maximum enzyme activity, $K_m$ represents the half-saturation constant or the substrate concentration at which the reaction rate equals $V_{max}/2$ and $S$ is the substrate amount. Both the $V_{max}$ and $K_m$ parameters were approximated by the Michaelis–Menten Equation (1) with the nonlinear regression routine of STATISTICA.

*2.5. Statistics*

All statistical analyses and graph works were performed using R software (version 1.4.1106) [27]. The measured variables were analyzed using a linear model, with the pasture type, N fertilization rate and interaction term as the fixed factors. In each case, the adequacy of the model was assessed by examining the appropriateness of residual plots. The effective means were separated at $p < 0.05$ with Tukey's post hoc tests, using the "lsmeans" function of the "multcomp" package [28]. Spearman's correlation and linear regression tests were performed to establish the relationships among the measured variables. Correlation coefficients between variables and their significance were computed using the "sjPlot" package by the pairwise deletion procedure. The "lavaan" package in R [29] was used to perform structural equation modeling (SEM) to assess the measured plant traits' direct and indirect controls on $N_2O$ emissions. A conceptual model based on the hypotheses and theoretical knowledge of plant effects on $N_2O$ emissions was constructed. Before the SEM analyses, the predictor and dependent parameter units were adjusted to obtain comparable parameter variances. Nonsignificant relationships were removed following a stepwise procedure using model modification indices [30]. The quality of the SEM model was determined using the chi-square goodness-of-fit statistic ($p > 0.05$, which indicated a statistically significant model fit), the root means square error of approximation value (RMSEA < 0.08), the comparative fit index (CFI > 0.95) and the standardized root mean square residual (SRMR < 0.08).

## 3. Results

### 3.1. Biomass Allocation and CN Partitioning

Generally, N application increased the DM yields, the C and N yields and reduced root/shoot and BGB C/N ratios significantly ($p < 0.001$). For instance, N application increased the total biomass yield by approximately four-fold (Table 1) when compared with the control treatment ($p < 0.05$). However, the N150 and N300 treatments were not significantly different, particularly in harvested biomass, stubble, roots, total DM or C yields, except for AGB C in the binary mixtures. At the third harvest, the *PL* DM yield increased with the N rate, where it was two-fold higher for N300 (Appendix A, Table A2). In addition, the fertilizer application at N150 or N300 reduced the root/shoot ratio in the *LP* swards and *PL* swards, with no effect ($p > 0.05$) on the mixed swards (Table 1). A fertilizer response curve (Figure 1a) showed that the swards could achieve the maximum DM yield at N rates less than 300 kg ha$^{-1}$ (~240 kg ha$^{-1}$).

The sward differences significantly influenced ($p < 0.05$) the DM yields (except harvested biomass), N and C yields (except AGB N) and root/shoot and C/N ratios, but significant sward x fertilizer rate interaction terms were evident (Table 1). Thus, the binary mixture achieved comparable yields to the *LP* monoculture but was around 50% higher than the *PL* monoculture independent of the fertilizer rate ($p < 0.05$; Table 1 and Figure 1a). The proportion of *PL* in the binary mixtures ranged between 7 and 26% (mean = 13%), depending on the N rate or growth stage (Appendix A, Figure A1). Generally, the *PL* proportion was reduced in the N-treated soils compared with the control, but the differences were only significant for the second and third harvests (Appendix A, Figure A1). On the other hand, the *PL* proportions increased with the increasing growth stage in the control plots but not the fertilized plots, though the differences between the second and third harvests were statistically insignificant ($p > 0.05$).

**Table 1.** Dry matter yield, C:N ratios and N uptake/use efficiencies from *Lolium perenne* (*LP*) and *Plantago lancelota* (*PL*) swards as affected by different fertilization rates.

| Nitrogen Rate (N) | 0 N | | | 150 N | | | 300 N | | | *p*-Value | | |
|---|---|---|---|---|---|---|---|---|---|---|---|---|
| Pasture (P) | *LP* | *LP-PL* | *PL* | *LP* | *LP-PL* | *PL* | *LP* | *LP-PL* | *PL* | P | N | P × N |
| Species proportion (%) | 100 | 74.4/ 25.6 | 100 | 100 | 93.3/6.7 | 100 | 100 | 91.9/ 8.1 | 100 | - | - | - |
| **DM yield (g m$^{-2}$)** | | | | | | | | | | | | |
| Harvested | 53.9 [A] (4.15) | 49.3 [A] (7.76) | 46.9 [A] (3.84) | 301.6 [B] (28.8) | 296.5 [B] (24.1) | 182.2 [B] (23.0) | 351.5 [B] (23.2) | 385.0 [B] (36.7) | 270.2 [B] (12.7) | ns | *** | ** |
| Stubble | 33.4 [aA] (1.95) | 31.6 [aB] (2.52) | 15.9 [bA] (1.26) | 94.0 [aB] (4.72) | 88.3 [aB] (3.96) | 48.1 [bB] (7.18) | 118.9 [B] (15.6) | 100.7 [B] (12.8) | 64.6 [B] (2.98) | *** | *** | ** |
| Roots | 59.7 [A] (6.35) | 58.7 [A] (8.09) | 33.2 [A] (2.65) | 164.3 [B] (19.1) | 147.9 [B] (11.7) | 95.5 [B] (15.1) | 187.3 [bB] (16.1) | 185.9 [bB] (13.5) | 88.4 [aB] (9.44) | *** | *** | ** |
| Total | 147.0 [aA] (4.3) | 139.5 [aA] (16.2) | 96.0 [bA] (4.1) | 559.9 [aB] (45.7) | 532.7 [aB] (17.0) | 325.8 [bB] (44.1) | 657.7 [aB] (44.7) | 671.7 [aB] (49.9) | 423.2 [bB] (16.1) | *** | *** | ** |
| Root/shoot ratio | 0.69 [A] (0.09) | 0.42 (0.03) | 0.54 [A] (0.07) | 0.42 [B] (0.03) | 0.39 (0.05) | 0.41 [A] (0.03) | 0.40 [aB] (0.04) | 0.39 [a] (0.03) | 0.26 [bB] (0.03) | * | *** | ns |
| **N and C yields (g m$^{-2}$)** | | | | | | | | | | | | |
| AGB N | 1.15 [A] (0.02) | 1.06 [A] (0.09) | 1.22 [A] (0.08) | 5.24 [B] (0.53) | 6.06 [B] (0.14) | 4.90 [B] (0.82) | 8.90 [C] (0.62) | 9.17 [C] (0.77) | 9.30 [C] (0.37) | ns | *** | ns |
| BGB N | 0.52 [aA] (0.04) | 0.56 [aA] (0.07) | 0.37 [aA] (0.02) | 1.32 [aB] (0.12) | 1.39 [bB] (0.08) | 0.83 [bB] (0.14) | 1.80 [aC] (0.14) | 1.89 [bC] (0.11) | 1.14 [bB] (0.10) | *** | *** | * |
| AGB C | 34.1 [bA] (1.0) | 33.2 [abA] (3.3) | 24.4 [aA] (1.6) | 164.9 [bB] (13.4) | 151.8 [bB] (10.8) | 94.5 [aB] (12.6) | 197.1 [bB] (15.9) | 204.1 [bC] (16.6) | 139.2 [aB] (5.4) | *** | *** | ** |
| BGB C | 22.9 [bA] (1.5) | 24.4 [bA] (3.2) | 13.4 [aA] (1.1) | 60.4 [bB] (4.2) | 60.0 [bB] (4.7) | 34.5 [aB] (5.8) | 70.9 [bB] (4.1) | 70.0 [bB] (5.0) | 33.6 [aB] (3.4) | *** | *** | ** |
| BGB C/N ratio | 44.6 [bAB] (3.1) | 46.0 (1.5) | 35.8 [aAB] (0.8) | 46.0 [A] (1.5) | 43.1 (1.3) | 41.4 [B] (1.4) | 39.6 [Bb] (1.1) | 37.2 [b] (2.2) | 29.5 [aA] (2.2) | ** | *** | ns |

*LP* represents *Lolium perenne*; *PL* represents *Plantago lanceolata*. SEM = standard error of the mean (pooled across pastures) and fertilization rates; means with different letters are different at $p < 0.05$, [ab] represent the effect of the sward type and [ABC] represent the effect of the fertilization rate; [ns] $p > 0.05$, * $p < 0.05$, ** $p < 0.01$ and *** $p < 0.001$. AG is the aboveground biomass (harvested + stubbles); BG is thebelowground biomass. The plant biomass$^{\alpha}$ is AG + BG.

*Plantago lanceolata* had a lower root/shoot ratio than the *LP* treatment (Table 1). The nitrogen yield belowground was higher for the *PL* or *LP-PL* swards than *LP* at N150 and N300 rates. Like AGB, the N yield increased with the N rate in BGB, except for *PL*, which showed no significant differences between the N150 and N300 rates. A regression analysis (Figure 1b) showed that the total herbage N yield increase was greater per gram of plantain DM yield ($1.96 \cdot 10^{-2}$ g g$^{-1}$) than per gram of *LP* DM yield ($1.31 \cdot 10^{-2}$ g g$^{-1}$) or the mixed DM yield ($1.41 \cdot 10^{-2}$ g g$^{-1}$), respectively. Moreover, C yields both above- and belowground were generally lower for *PL* swards than *LP* or the binary swards. The C/N ratio of the BGB ranged between 30 and 46 and was higher for the *LP*-rich swards than the *PL* monoculture (Table 1).

### 3.2. Microbial Activity (LAP Expression)

LAP expression in the soils increased at all the N rates after the first growth stage and appeared to stabilize or drop slightly at the third growth stage (Figure 2a). Increasing the N availability in the soils significantly ($p < 0.05$) reduced the LAP activity at each growth stage. The LAP activity was highest in the mixed sward and lowest in *PL* monoculture swards ($p < 0.05$), except for N150, where the differences between the two monoculture swards were marginal (Figure 2b). The Regression analysis (Figure 3) showed a significant relationship (quadratic) between the microbial enzyme expression and belowground biomass C/N, where higher C yields relative to N belowground are generally associated with more elevated microbial enzyme (LAP) activity.

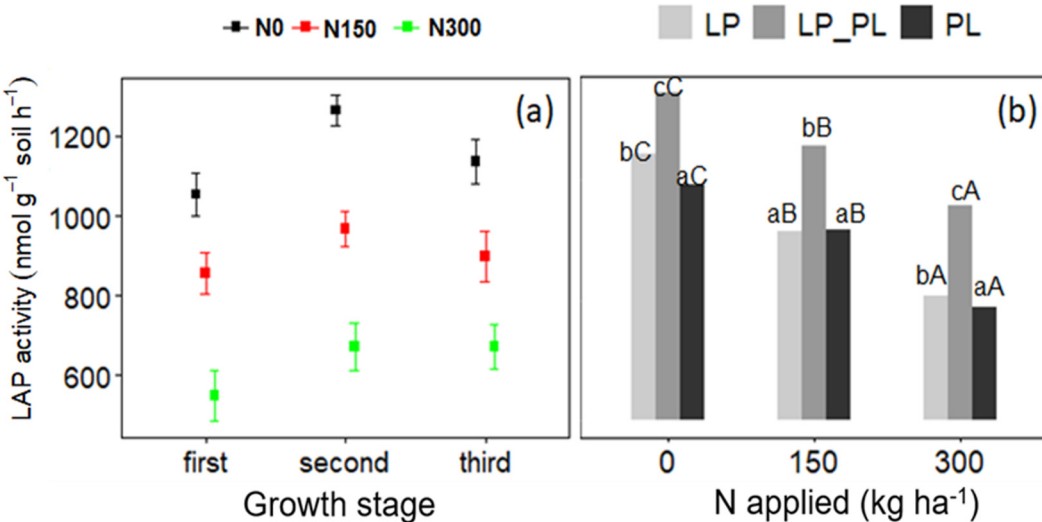

**Figure 2.** Graphs showing leucine aminopeptidase (LAP) expression in fertilized soils cultivated with forage species as affected by (**a**) the stage of growth and (**b**) sward type. Lowercase and uppercase letters represent significant differences between the sward types and the N rate, respectively. *LP* represents *Lolium perenne*; *PL* represents *Plantago lanceolata*.

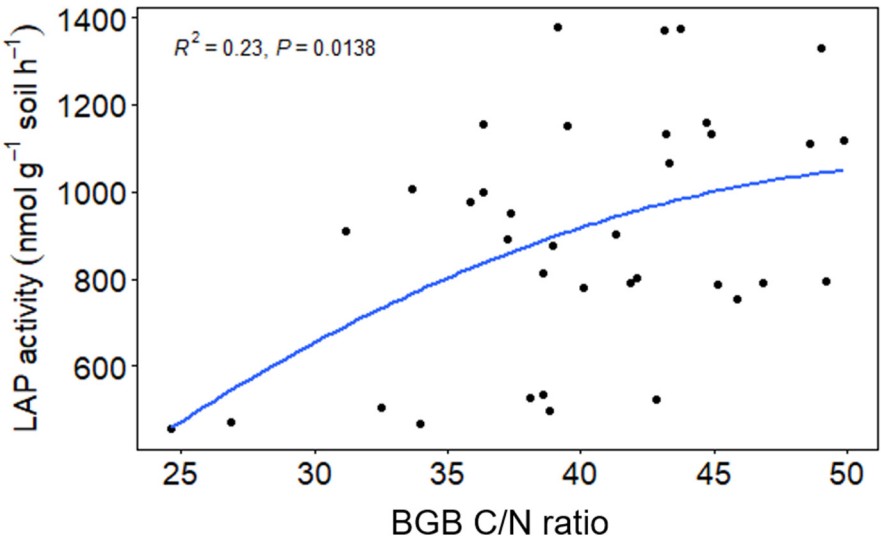

**Figure 3.** A quadratic response curve showing the relationship between the microbial enzyme (LAP) activity and belowground biomass C/N ratio.

### 3.3. Nitrous Oxide Emission

The daily $N_2O$ emission ranged from 0 to 8.70 mg N $m^{-2}$ across all the treatments, with means ($\pm$SD) of $0.01 \pm 0.02$, $0.40 \pm 0.49$ and $1.37 \pm 1.94$ mg N $m^{-2}$ at the N0, N150 and N300 rates, respectively. Peak fluxes occurred a few days after each fertilizer application event. The highest peak fluxes of 0.11, 1.67 and 8.70 mg N $m^{-2}$ $d^{-1}$ observed for the N0, N150 and N300 rates were recorded from the *PL* swards (Figure 4). The maximum emission peak occurred 17, 3 and 2 days after the first, second and third fertilizer applications, respectively. Generally, increasing the N rates increased cumulative $N_2O$ emission (up to about 85-fold), but the differences were insignificant in some cases (Table 2). For example, whereas the N300 rate consistently increased $N_2O$ emissions across the swards compared with the control, N addition at N150 did not significantly increase $N_2O$ emissions from the *LP* swards ($p > 0.05$; Table 2). Unlike the monocultures, the N150 and N300 rates did not significantly affect ($p > 0.05$) cumulative $N_2O$ emissions per $m^2$, per root biomass or total biomass in the mixed swards (Table 2). However, $N_2ON$ EF was two-fold higher for N300 than the N150 rates. Moreover, the sward effect was significant on $N_2ON$ EF and was

higher for *PL* swards relative to *LP* at the N150 rate and relative to the binary swards at the N300 rate (Table 2). The sward effects on N uptake, N surplus, residual soil N, AGB or whole plant NUE were not significant (Table 2); however, BGB NUE was lower for the *PL* swards than *LP* or *LP-PL* (Appendix A, Figure A2). With increasing N fertilizer, plant N uptake and N surplus increased across all the treatments (Table 2).

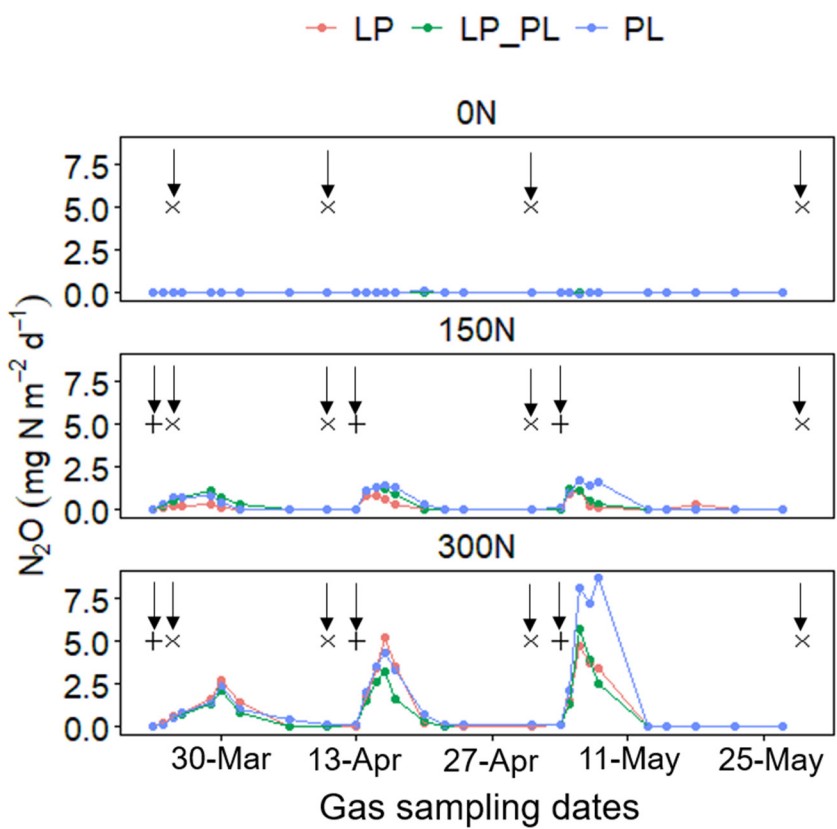

**Figure 4.** Graphs showing daily $N_2O$ fluxes from the diverse swards at different fertilization rates. *LP* represents *Lolium perenne*; *PL* represents *Plantago lanceolate*. Arrows with + represent fertilizer application events, and arrows with x represent a biomass harvesting event.

### 3.4. Relationships between the Measured Variables

Nitrous oxide-N emissions did not significantly correlate with any plant/microbial trait in the N0 treatment (Table 3). At N150, lower $N_2O$-N emissions were significantly ($p < 0.05$) associated with a higher BG C/N ratio and higher AG NUE; however, higher $N_2O$-N emissions were associated with higher *PL* proportions. Lower $N_2O$-N emission was associated with higher BG DM and higher microbial activity at the N300 rate (Table 3). Generally, a higher percentage of *PL* in the sward correlated negatively with BG DM, BG C/N ratio and AG NUE at all the N rates and BG C/N ratio correlated positively with BG NUE at all the N rates.

The structural equation models (SEM) shown in Figure 5 describe the direct and indirect effects of the measured plant/microbial traits on $N_2O$ emissions. Accordingly, the $N_2O$ emissions from the N150 treatment were predominantly positively influenced by the percentage of *PL* in the sward ($r = 0.50$) and negatively influenced by the C/N ratio of the belowground biomass ($r = -0.42$) and the root/shoot ratio ($r = -0.30$), with all the contributing factors resulting in a combined coefficient of determination of $R^2 = 0.62$ (Figure 5). While the *PL* proportion negatively influenced both the LAP activity ($r = -0.68$, $R^2 = 0.38$) and the whole plant N uptake ($r = -0.44$, $R^2 = 0.21$), this did not translate into a strong influence on $N_2O$ emission. For N300, the percentage of *PL* was again positively linked to $N_2O$ emissions ($r = 0.68$), with no other factor having a strong direct impact.

Besides, higher *PL* proportions negatively impacted LAP activity ($r = -0.89$, $R^2 = 0.34$); this, again, did not translate into a strong impact on $N_2O$ emissions ($r = -0.22$). At both N rates, higher *PL* proportions reduced LAP expression and BG C/N ratio. The impact of the plant and microbial factors on $N_2O$ emission was reduced by about 10% when the N application increased by 100%. Bivariate regression models based on pooled data across soil N rates showed the $N_2O$ emission-reducing effect of microbial activity and sward traits, with microbial activity, total biomass yield, root biomass yield, C yield and C/N ratio being significant (Table 4). However, the biomass C yield belowground showed the greatest impact on $N_2O$ emission from the swards (Table 4).

**Table 2.** Cumulative $N_2O$ emissions (83 days), as well as $N_2O$ emissions in relation to above- (AG) and belowground biomass (BG) among the different treatments.

| Nitrogen Rate (N) | 0 N | | | 150 N | | | 300 N | | | *p*-Value | | |
|---|---|---|---|---|---|---|---|---|---|---|---|---|
| Pasture (P) | LP | LP-PL | PL | LP | LP-PL | PL | LP | LP-PL | PL | P | N | P × N |
| | | | | | $N_2O$-N emission | | | | | | | |
| $N_2O$, g N m$^{-2}$ | 0.69 [aA] (0.2) | 0.71 [aA] (0.1) | 0.82 [aA] (0.3) | 9.23 [aA] (1.6) | 16.9 [abB] (4.2) | 23.6 [bB] (4.3) | 55.3 [abB] (7.9) | 43.2 [aB] (8.5) | 85.4 [bC] (13.1) | ns | *** | ** |
| $N_2O$, g N BGB$^{-1}$ | 0.01 [aA] (0.0) | 0.01 [aA] (0.0) | 0.02 [aA] (0.0) | 0.06 [aA] (0.0) | 0.12 [abA] (0.0) | 0.28 [bB] (0.1) | 0.30 [aB] (0.0) | 0.24 [aA] (0.1) | 0.99 [bC] (0.2) | *** | *** | *** |
| $N_2O$, g N total DM$^{-1}$ | 0.01 [aA] (0.0) | 0.01 [aA] (0.0) | 0.01 [aA] (0.0) | 0.02 [aA] (0.0) | 0.03 [abA] (0.0) | 0.08 [bB] (0.0) | 0.08 [aB] (0.0) | 0.07 [aA] (0.0) | 0.20 [bC] (0.0) | ns | *** | ** |
| $N_2O$, g N g Nuptake$^{-1}$ | 0.41 [aA] (0.1) | 0.46 [aA] (0.1) | 0.49 [aA] (0.2) | 1.37 [aA] (0.1) | 2.30 [abB] (0.6) | 4.59 [bB] (1.5) | 5.11 [abB] (0.5) | 4.06 [aB] (1.0) | 8.32 [bB] (1.6) | * | *** | ns |
| $N_2O$-N EF, % | - | - | - | 0.06 [aA] (0.01) | 0.11 [ab] (0.03) | 0.15 [bA] (0.03) | 0.18 [abB] (0.03) | 0.14 [a] (0.03) | 0.28 [bB] (0.04) | ** | *** | ns |
| *N use* | | | | | | | | | | | | |
| N uptake, (kg ha$^{-1}$ | 16.7 [A] (0.2) | 65.6 [B] (6.0) | 15.9 [A] (0.8) | 65.6 [B] (6.0) | 74.5 [B] (2.0) | 57.4 [B] (9.4) | 107.0 [C] (6.7) | 110.6 [C] (8.3) | 104.4 [C] (3.7) | ns | *** | ns |
| N surplus, kg ha$^{-1}$ | −16.7 [A] (0.2) | −16.2 [A] (1.6) | −15.9 [A] (0.8) | 84.4 [B] (6.0) | 75.6 [B] (2.0) | 92.6 [B] (9.4) | 193.0 [C] (6.7) | 189.4 [C] (8.3) | 195.6 [C] (3.7) | ns | *** | ns |
| Soil residual N, % | 0.12 (0.001) | 0.12 (0.002) | 0.12 (0.001) | 0.12 (0.003) | 0.12 (0.001) | 0.13 (0.003) | 0.12 (0.003) | 0.09 (0.029) | 0.12 (0.002) | ns | ns | ns |
| AGB NUE, g g$^{-1}$ | - | - | - | 0.27 (0.03) | 0.33 (0.01) | 0.25 (0.06) | 0.26 (0.02) | 0.27 (0.03) | 0.27 (0.01) | ns | ns | ns |
| BGB NUE, g g$^{-1}$ | - | - | - | 0.05 (0.01) | 0.06 (0.01) | 0.03 (0.01) | 0.04 (0.006) | 0.04 (0.006) | 0.03 (0.003) | * | ns | ns |
| Whole plant NUE, g g$^{-1}$ | - | - | - | 0.33 (0.04) | 0.39 (0.02) | 0.28 (0.07) | 0.30 (0.23) | 0.32 (0.03) | 0.30 (0.09) | ns | ns | ns |

Values in parenthesis are the standard error of the mean; means with different letters are different at $p < 0.05$, [ab] represents the effect of sward type and [ABC] represents the effect of the fertilization rate; [ns] $p > 0.05$, * $p < 0.05$, ** $p < 0.01$ and *** $p < 0.001$. *LP* represents *Lolium perenne*; *PL* represents *Plantago lanceolata*.

**Table 3.** Spearman's correlation [α] coefficients showing the relationships between the $N_2ON$ emission, biomass variables and microbial activity.

| | BG DM | AG C/N | BG C/N | NUpE | $N_2ON$ | BG/AG | BG NUE | AG NUE | LAP | Plan |
|---|---|---|---|---|---|---|---|---|---|---|
| | | | | | *N0* | | | | | |
| AG DM | 0.75 ** | 0.75 ** | 0.54 | 0.64 * | 0.00 | 0.22 | 0.41 | 0.72 * | 0.54 | −0.78 ** |
| BG DM | | 0.71 * | 0.76 ** | 0.40 | −0.28 | 0.78 ** | 0.78 ** | 0.64 * | 0.53 | −0.79 ** |

**Table 3.** *Cont.*

|  | BG DM | AG C/N | BG C/N | NUpE | N$_2$ON | BG/AG | BG NUE | AG NUE | LAP | Plan |
|---|---|---|---|---|---|---|---|---|---|---|
| AG C/N |  |  | 0.73 ** | 0.08 | −0.09 | 0.29 | 0.57 | 0.95 *** | 0.75 ** | −0.64 * |
| BG C/N |  |  |  | −0.01 | 0.06 | 0.54 | 0.93 *** | 0.74 ** | 0.50 | −0.69 * |
| NUpE |  |  |  |  | 0.11 | 0.15 | 0.01 | 0.03 | −0.06 | −0.40 |
| N$_2$ON |  |  |  |  |  | −0.34 | −0.15 | 0.00 | −0.19 | 0.20 |
| BG/AG |  |  |  |  |  |  | 0.69* | 0.22 | 0.26 | −0.44 |
| BG NUE |  |  |  |  |  |  |  | 0.53 | 0.34 | −0.61 * |
| AG NUE |  |  |  |  |  |  |  |  | 0.69 * | −0.73 ** |
| LAP |  |  |  |  |  |  |  |  |  | −0.46 |
| | | | | | *N150* | | | | | |
| AG DM | 0.71 * | 0.59 * | 0.46 | 0.48 | −0.49 | −0.23 | 0.08 | 0.66 * | 0.22 | −0.73 ** |
| BG DM |  | 0.37 | 0.36 | 0.62 * | −0.56 | 0.47 | 0.31 | 0.53 | 0.16 | −0.76 ** |
| AG C/N |  |  | 0.48 | −0.30 | −0.60 * | −0.18 | 0.28 | 0.95 *** | −0.09 | −0.81 ** |
| BG C/N |  |  |  | 0.15 | −0.59 * | −0.11 | 0.62 * | 0.52 | −0.27 | −0.65 * |
| NUpE |  |  |  |  | −0.03 | 0.20 | −0.06 | −0.15 | 0.23 | −0.19 |
| N$_2$ON |  |  |  |  |  | −0.19 | −0.38 | −0.73 * | −0.27 | 0.73 ** |
| BG/AG |  |  |  |  |  |  | 0.37 | −0.08 | −0.06 | −0.15 |
| BG NUE |  |  |  |  |  |  |  | 0.25 | −0.62 * | −0.40 |
| AG NUE |  |  |  |  |  |  |  |  | 0.10 | −0.87 *** |
| LAP |  |  |  |  |  |  |  |  |  | −0.04 |
| | | | | | *N300* | | | | | |
| AG DM | 0.83 ** | 0.80 ** | 0.69 * | 0.66 * | −0.53 | 0.48 | 0.86 *** | 0.80 ** | 0.50 | −0.60 * |
| BG DM |  | 0.70 * | 0.53 | 0.41 | −0.60 * | 0.84 ** | 0.82 ** | 0.71 * | 0.59 * | −0.65 * |
| AB C/N |  |  | 0.83 ** | 0.17 | −0.52 | 0.50 | 0.89 *** | 0.99 *** | 0.58 | −0.78 ** |
| BG C/N |  |  |  | 0.10 | −0.57 | 0.38 | 0.81 ** | 0.83 ** | 0.41 | −0.67 * |
| NUpE |  |  |  |  | −0.07 | −0.07 | 0.36 | 0.19 | −0.02 | 0.11 |
| N$_2$ON |  |  |  |  |  | −0.54 | −0.43 | −0.52 | −0.69 * | 0.45 |
| BG/AG |  |  |  |  |  |  | 0.59 * | 0.48 | 0.54 | −0.72 ** |
| BG NUE |  |  |  |  |  |  |  | 0.89 *** | 0.37 | −0.75 ** |
| AG NUE |  |  |  |  |  |  |  |  | 0.55 | −0.79 ** |
| LAP |  |  |  |  |  |  |  |  |  | −0.44 |

[α] Computed correlation used Spearman's method with pairwise deletion. BG—belowground biomass; AG—aboveground biomass; DM—dry matter; C—carbon; N—nitrogen; RNUE/SNUE—root/shoot nitrogen use efficiency; NUpE—whole plant N uptake; LAP—leucine aminopeptidase, a proxy for microbial activity; level of significance: *** $p < 0.001$, ** $p < 0.01$, * $p < 0.05$.

**Table 4.** Linear effect of the measured biomass traits on N$_2$O emissions.

| Parameter | Estimate | *F*-Value | *p*-Value | $R^2$ |
|---|---|---|---|---|
| N surplus (g m$^{-2}$) | 0.68 | 2.46 | ns | 0.20 |
| N uptake (g m$^{-2}$) | −6.76 | 2.56 | ns | 0.20 |
| LAP activity | −0.03 | 3.94 | ns | 0.28 |
| LAP activityˆ2 | −1.39 | 11.98 | ** | 0.73 |
| DM production | −0.07 | 9.88 | * | 0.50 |
| Root biomass | −0.21 | 10.01 | * | 0.50 |
| Root/shoot ratio | −70.0 | 2.55 | ns | 0.20 |
| BGB C | −0.61 | 15.74 | ** | 0.61 |
| BGB C/N ratio | −1.40 | 5.68 | * | 0.36 |
| NUE | −89.76 | 1.29 | ns | 0.11 |
| ABG NUE | −79.88 | 0.55 | ns | 0.05 |
| BGB NUE | −493.41 | 4.41 | ns | 0.31 |

BGB—below ground biomass; AGB—aboveground biomass; NUE—nitrogen use efficiency; LAP—Leucine aminopeptidase; LAP—leucine aminopeptidase, a proxy for microbial activity; level of significance: ** $p < 0.01$, * $p < 0.05$.

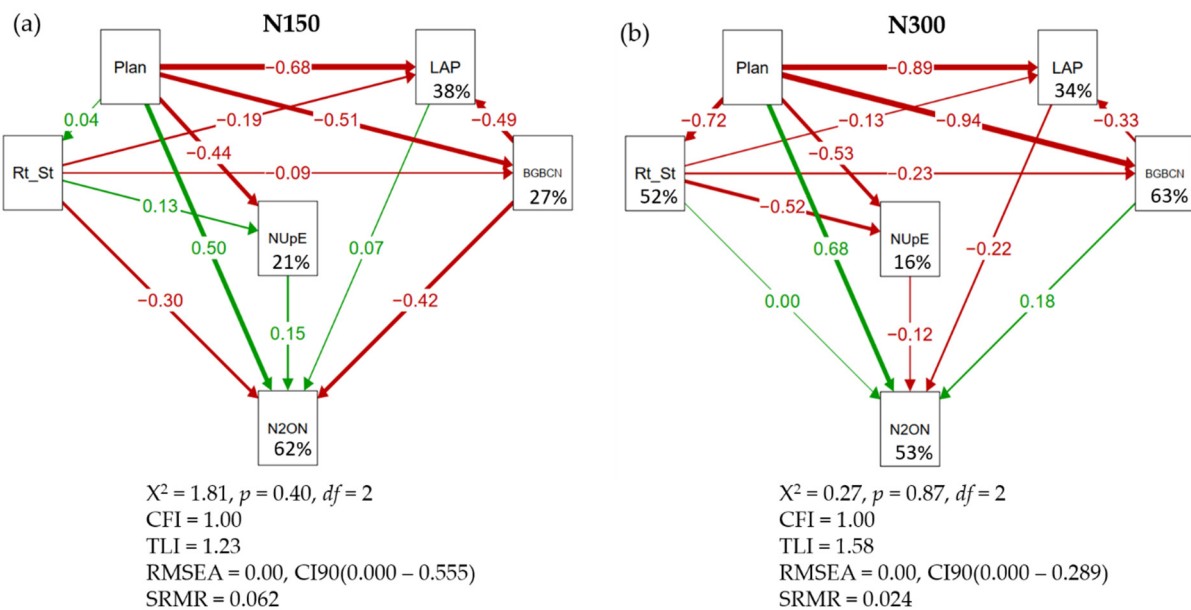

**Figure 5.** Structural equation models showing the relationships between the measured variables at (**A**) the N150 fertilizer rate and (**B**) at the N300 fertilizer rate. Plan–forage plantain proportion; NUpE—whole plant N uptake; Rt_St—root/shoot ratio; BGBCN—belowground biomass C/N ratio; LAP—leucine aminopeptidase, a proxy for microbial activity; green arrows represent a positive association, and red arrows represent a negative association; numbers within arrows represent standardized path coefficients, whereby negative path coefficients indicate a negative relationship. The higher the absolute value of the path coefficient, the more important is the path. Numbers in the white boxes show the percentage of explained variance of each of the endogenous variables. $X^2$ indicates the fit of the model, Comparative Fit Index (CFI), Tucker Lewis Index (TLI), and the Root Mean Square Error of Approximation (RMSEA).

## 4. Discussion

### 4.1. Dry Matter, Carbon and Nitrogen Yields

Nitrogen application generally increased dry matter, C and N yields; thus, the soils overcame nutrient limitations, as seen in previous pasture N fertilizer studies [31,32]. We expected this trend as N is a major component of chlorophyll required for plant metabolism, including proteins and carbohydrate synthesis [33]. However, DM yields did not differ significantly between the two fertilizer rates across the swards, despite a general tendency to increase the yields [34]. Indeed, the regression analysis showed that the maximum DM yield was attained at an N rate of about 240 kg ha$^{-1}$ (Figure 1a); thus, it appears the additional N beyond this rate contributed only marginally to the DM yield. This situation might explain the relatively high N surplus and N$_2$O emissions observed for the N300 treatments compared with N150 in the current study (Table 2). In any case, N addition stimulated the root growth and development, enhancing the uptake of other nutrients [35]. Generally, more N was taken up as the N rate increased; thus, N fertilization reduced the C/N ratio in both below- and aboveground biomass, with this trend being most evident in the N300 treatment, indicating a luxury uptake of N, similar to earlier reports in maize straw [36] and grass swards [37].

The response of the *PL* sward to the N application was weaker than the *LP*-rich swards (Figure 1a); however, the N concentration and the rate of increase in N yield per DM increase were higher for the *PL* swards than the *LP*-rich swards (Figure 1b), similar to reports by previous authors [38]. Furthermore, *LP* and *PL* differ in their preference for NH$_4^+$ vs. NO$_3^-$, with *LP* and *PL* preferring NH$_4^+$ and NO$_3^-$, respectively [39,40]. We did not measure soil pH changes in the current study; however, urea application is known to cause large pH swings, increasing to about 8.0 and then tracked downwards over time due to the nitrification and denitrification of NO$_3^-$ reversing the trend a little [41].

Therefore, N application could have probably led to large pH swings favoring N uptake by *PL* swards [42], leading to higher N yields.

It was hypothesized that the mixed sward would yield higher DM than the mono-species swards at all N rates due to their higher capacity to utilize resources efficiently [43]. However, the DM yield was slightly higher (about 3%) for the mixed swards than the *LP* (Table 1). The statistically similar DM yield observed for mixed and monoculture *LP* swards might be due to the relatively low proportion of *PL* (average of about 13%) in the mixed swards, which probably did not suffice for the optimal niche utilization. The higher DM yield from *LP* compared with *PL* swards was also reported in a meta-analysis [44] and attributed to the high water content in *PL.* Additionally, *PL*'s lower root/shoot ratio compared to the *LP* mono-crop sward might be due to their low nutrient or light acquisition capacity [45].

The proportion of *PL* in the mixed swards increased over time, particularly under low soil N conditions (Appendix A, Figure A1). This observation was comparable to reports by Olff and Bakker [46], who observed increased *PL* proportions in mixed swards on nutrient-poor soils significantly when the cutting intervals were extended. However, *PL* proportions remained low in the mixed swards under medium-to-high N fertilization, suggesting competition from *LP*. This trend was expected, as the performance of individual species in mixtures depends on the companion species' interspecific interactions, which can be synergistic or antagonistic. The strength of the interaction also depends on the relative abundances of the species involved [47]. Perennial ryegrass is a fast and aggressive species and has a high capacity to utilize N for biomass productivity, as shown in this study, and might dominate mixed plant communities [12]. This attribute of *LP* swards might partly explain their higher response to N fertilization than the *PL* swards (Figure 1a).

### 4.2. Microbial Activity

The swards' structural and functional differences affected LAP expression in the soils, which reflected in a higher enzyme activity associated with the mixed swards relative to the monoculture swards at all N fertilization rates. This trend might be due to the high exudation associated with mixed-species communities and the consequent stimulation of microbial activity [48]. Root exudates are the primary source from which soil microbes derive energy for biomass production [49]. Furthermore, it was found that a positive association existed between LAP expression and BGB C/N ratio (Figure 3). A positive association between increased root biomass, root exudates and microbial biomass and increased species richness has been previously reported [50,51]. Prommer et al. [52] reported a direct relationship between root biomass, microbial growth and microbial biomass carbon. Thus, a higher C availability belowground might have stimulated microbial growth [53]. This observation might explain why there appeared to be an increased LAP activity associated with the increasing growth stage and the mixed swards (Figure 2a). However, soil microbial communities can also affect the amount and composition of root exudates [54]. Sørensen et al. [55] contended that various plant organic compounds affect the structure, abundance and microbial activity. Thus, more diverse exudates would stimulate a higher microbial diversity [56] and alter extracellular enzyme expression.

As we hypothesized, we found a decreasing LAP expression associated with increasing N fertilization (Figure 2b). This trend was also reported for the temperate Mongolian steppe [57] and other grassland ecosystems [17,58]. Evolutionary–economic mechanisms of enzyme production have been suggested across a range of ecosystems [59]. This means that microbial communities can be considered economic units that maximize their productivity by allocating resources to extracellular nutrient-releasing enzymes, depending on nutrient limitation. Hence, N deficiencies stimulate the production of respective acquiring enzymes while N fertilization suppresses the expression of N-degrading enzymes. Besides, the LAP expression as a response to N availability might be mediated by environmental conditions, including the soil pH, soil water availability and enzyme co-factors availability [57]. Although we did not measure the soil pH changes in this study, it is speculated that the N

addition must have reduced soil pH [60]. Furthermore, N addition is also associated with an increased biomass production (Table 1), which increases leaf transpiration, leading to water loss [61]. Thus, the potentially low soil pH and reduced water availability due to the increasing N addition may have negatively affected microbial activity [62].

*4.3. Nitrous Oxide Emissions from the Swards*

It was hypothesized that higher *PL* proportions in the swards could reduce $N_2O$ emissions from the soils, as *PL* might release secondary plant metabolites (PSM), including mucilage, pectin flavonoids and tannins and inhibit microbial growth [63]. Furthermore, these chemicals, released via root exudates, inhibit nitrification in soils [13,64]. However, contrary to our hypothesis, a higher *PL* proportion in the swards resulted in higher $N_2O$ emissions (Table 2 and Figure 5). Moreover, BGB NUE was lower for the *PL* swards than *LP* or mixed swards (Figure 5). Besides, $N_2O$ emissions were still lower for the *LP*-rich swards than the *PL* mono-crop sward when expressed as a yield-scaled $N_2O$ emission or $N_2O$ emission intensity (cumulative $N_2O$ divided by biomass or N uptake). This expression is to identify potential trade-offs between the biomass production and environmental sustainability [9].

Nevertheless, the relatively high $N_2O$ emission observed for the *PL* mono-crop sward in our study is consistent with a previous study that reported no difference in $N_2O$ emission or the growth of ammonium-oxidizing bacteria (AOB) when ruminant urine N was applied to ryegrass–white clover pasture and pastures containing 30–100% *PL* [6]. In that study, there was no evidence of plantain affecting the soil inorganic N concentrations via biological nitrification inhibition following ruminant urine N application [26]. Although the presence of BNI compounds in root exudates was not determined in the current study, there was a negative relationship between *PL* proportion in the swards and microbial enzyme activity (Figure 3), suggesting a possible BNI effect from *PL* root exudates. In contrast to the current study, Carlton et al. [64] reported a reduced AOB abundance for a 2-year-old *PL*-rich pasture (ca. 30% *PL*), with a strong impact of contrasting soils and environmental conditions. The age effect appears to explain the lack of a plantain nitrification inhibition effect in the current study and previous mesocosm experiments [25,26], which involved less-established or young *PL* swards compared with the experiment by Carlton et al. [64]. Fuchs and Bowers [65] showed that the aucubin and catalpol concentrations in the *PL* swards increased significantly in the first months after sowing. Additionally, a symbiotic association between plants and arbuscular mycorrhizal fungi (AMF), which affects N cycling by reducing available N, is known to vary with plant age, with well-established AMF communities reducing the $N_2O$ emission significantly [66]. In the current experiment, the *PL* plants were probably too young to produce significant NI-associated substances.

Mixed swards have a higher N use efficiency than mono-crops due to their high potential for niche differentiation and species complementarity [43]. Moreover, Lama et al. [21] also showed that higher plant species richness in pastures reduced gross N mineralization, with small herb presence increasing gross N immobilization. Thus, it was hypothesized that the mixed swards would have a higher N uptake, higher NUE and lower $N_2O$ emissions than the *LP* mono-crop swards. In the current study, only a slightly higher N uptake (6%) and NUE (9%) and a slightly lower cumulative $N_2O$-N emission (7%) were observed for the mixed swards compared with the *LP* mono-crop sward. The marginal impact of the mixed swards on $N_2O$ emission in the current study might be due to the high *LP* proportions in the mixed swards (Table 1), resulting in comparable DM and belowground C yields.

*4.4. Pasture Resource Acquisition Affects $N_2O$ Emission*

Although the mixed swards were associated with relatively high microbial activity compared with the *LP* mono-crop sward (Figure 2), $N_2O$ emission did not differ, suggesting the presence of other confounding factors. Nevertheless, our results showed a potential $N_2O$ reduction effect of higher microbial activity and allocation of biomass resources belowground (DM production and C acquisition). Abalos et al. [9] suggested high biomass

production and fast N uptake as the main pathways that plants impact $N_2O$ emissions. These authors mentioned specific leaf area (SLA) and root length density (RLD) as particularly relevant in determining the $N_2O$ mitigation potential of a plant. Although not significant, N uptake negatively impacted $N_2O$ emission in the current study. The specific leaf area and RLD were not measured in this study; however, the observed significant quadratic effect of aboveground biomass production and the observed effect of BGB NUE ($p = 0.06$) on $N_2O$ emission might be related to variations in SLA and RLD of the swards, especially as both parameters have been linked with the relative growth rate and specific nitrogen absorption rate of roots [67].

In this study, higher belowground C related negatively with $N_2O$ emission but positively with microbial activity (Figure 3 and Table 4). These results point to the assertion that belowground C availability enhances microbial activity, which could increase the consumption or immobilization of soil mineral N [8] and reduce $N_2O$ emission. Root C:N ratios were shown to influence the gross N mineralization rates in soil, with high root C:N ratios having a strong negative effect on gross N mineralization [68]. Thus, the higher belowground C observed for *LP*-rich swards compared to *PL* swards might have reduced N mineralization, partly explaining the relatively low $N_2O$ emissions associated with the *LP*-based swards (Table 2). As shown in this study (Figure 2b), the abundance of soil microbes declined with increasing N fertilization, as previously reported [69], which consequently reduced the $N_2O$ mitigation potential of the pastures (Figure 5). These results suggested that low N input systems may have a higher potential to enhance beneficial plant–microbe symbiotic associations with positive consequences for $N_2O$ mitigation, the ozone layer and the Earth's climate system.

As expected, increasing soil N availability increased $N_2O$ emission linearly, probably due to the oxidation of $NH_4$ and reduction of $NO_3$ [70]. Furthermore, the similarity of soil residual N and $N_2O$ fluxes between the control and N treatments at the end of the study showed the complete uptake of the applied N by the system. Ammonium-forming fertilizers (urea, in this case) undergo rapid nitrification to stimulate $N_2O$ production. Generally, fertilizer application increases $N_2O$ emissions within approximately 5–8 weeks [71]. Thus, the $N_2ON$ EFs reported in this study (0.06–0.28%; Table 2) could be considered valid and similar to the values reported earlier for the site where the soils were taken, when slurry and cow excreta were applied to *LP*-based pastures [7,72]. However, the $N_2ON$ EFs were half the IPCC [73] default of 0.77% for the agroecological zone.

## 5. Conclusions

This study does not support the hypothesis that *Plantago lanceolata* swards emit lower $N_2O$ than *LP*-rich swards through a potential nitrification-inhibiting effect, although the results showed a negative relationship between the *PL* proportion and microbial activity. However, the current study showed a link between plant species and $N_2O$ emissions, linking plant growth and C and N dynamics to $N_2O$ emissions. The C-acquisition rate of plants may play a significant role in soil N dynamics, not specifically the choice of species. Plantain in the current experiment had a low C-acquisition rate, and for this reason, the $N_2O$ emissions were not reduced compared to *LP* with a relative higher C-acquisition rate. Thus, the results supported our hypothesis that belowground biomass productivity and C yield significantly affect $N_2O$ emissions from pasture soils and appeared to be the primary mechanism by which the forages affected $N_2O$ emissions in this study. Forage plant breeders could explore this to enhance the capacity of intensively cultivated grasslands to mitigate N losses to the environment.

Besides, the *LP-PL* mixed swards had slightly higher N uptake and NUE and a lower cumulative $N_2O$ emission than the *LP* mono-crop swards, although not statistically significant, probably due to improper species proportions of swards but significantly associated with higher microbial activity. Moreover, our results showed a significant association of LAP expression (microbial activity) with $N_2O$ emission, with a higher microbial activity associating with lower $N_2O$ emission ($R^2 = 0.73$). Hence, we accepted

the hypothesis that LAP expression could be a potential biomarker for $N_2O$ emission from foraging systems. Our study highlighted the need to factor plant traits into models that predict $N_2O$ emission from soils and showed that a higher fertilizer N deposition could reduce the $N_2O$ emission potential of pastures.

**Author Contributions:** J.K.N., C.S.M. and T.R. conceptualized and designed the research experiment and the hypotheses; E.B.H., T.R. and C.S.M. developed the research protocols and implemented the studies; E.B.H. carried out the data collection; E.B.H. and B.S.R. performed the soil and plant analyses; J.K.N. compiled the data, developed the model codes and performed the data analysis; J.K.N. prepared the manuscript's initial draft, with contributions from all co-authors, to produce the final manuscript; B.S.R., C.S.M. and T.R. edited the manuscript and F.T. provided oversight and took leadership responsibility for the research activity planning and execution. All authors have read and agreed to the published version of the manuscript.

**Funding:** This research was funded by Deutscher Akademischer Austauschdienst—DAAD, grant number 57344816, and the APC was funded by the German Research Foundation (DFG) within the funding programme "Open Access Publizieren".

**Data Availability Statement:** The datasets generated and/or analyzed during the current study are available at https://doi.org/10.6084/m9.figshare.15138708.v1, last accessed on 1 September 2021.

**Acknowledgments:** We are grateful to Rita Kopp and Katrin Helmich of the Grassland and Forage Science/Organic Agriculture Department at the CAU for their immense technical assistance during the chemical analyses of the plants and gas samples, respectively. Furthermore, we are indebted to Christof Kluß for his immense assistance during the analyses of the data.

**Conflicts of Interest:** The authors declare no conflict of interest, and the funders had no role in the study's design, in the collection, analyses or interpretation of the data, in the writing of the manuscript or in the decision to publish the results.

## Appendix A

**Table A1.** Physical and chemical constitutions of the soil used for the experiment.

| | |
|---|---|
| Sand (%) | 60.0 |
| Silt (%) | 26.0 |
| Clay (%) | 14.0 |
| Total N (%) | 0.14 |
| Organic matter (%) | 2.0 |
| C/N ratio | 8.4 |
| pH | 7.2 |
| $P_2O_5$ (mg/kg) | 310.0 |
| $K_2O$ (mg/kg) | 120.0 |
| Mg (mg/kg) | 73.0 |
| Mn (mg/kg) | 72.0 |
| Cu (mg/kg) | 2.8 |
| B (mg/kg) | 0.4 |
| Zn (mg/kg) | 20.0 |
| Na (mg/kg) | 4.8 |

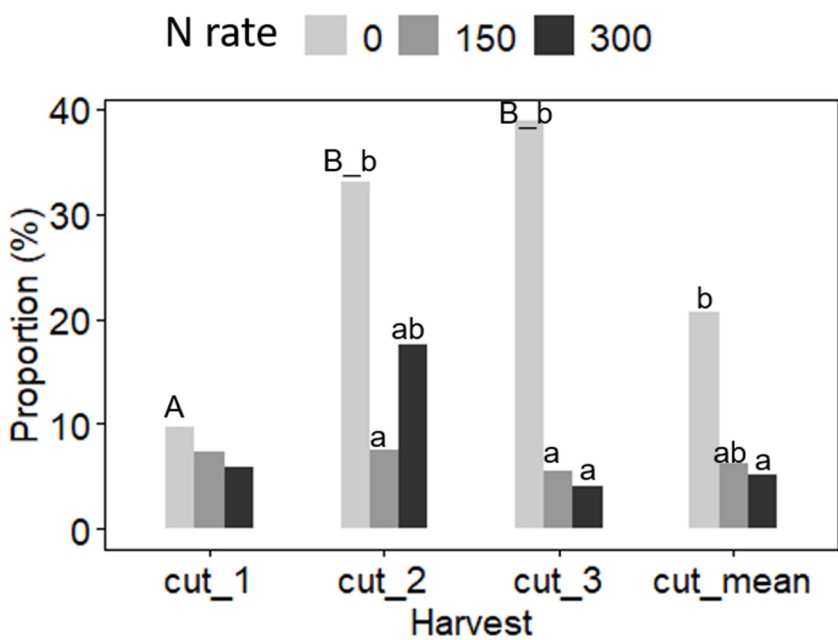

**Figure A1.** Plantago proportion in the perennial ryegrass—Plantago pasture mix (*LP-PL*) affected by fertilization and the growth/temporal cutting stage. Means with different letters are significantly different at $p < 0.05$, $^{AB}$ is the temporal cutting effect and $^{ab}$ the fertilization effect.

**Table A2.** Dry matter yield, C:N ratios and N uptake/use efficiencies from *Lolium perenne* (LP) and *Plantago lancelota* (PL) swards as affected by different fertilization rates.

| Pasture (P) | 0 N | | | 150 N | | | 300 N | | | F/p-Value | | |
|---|---|---|---|---|---|---|---|---|---|---|---|---|
| N Rate kg/ha | *LP* | *LP-PL* | *PL* | *LP* | *LP-PL* | *PL* | *LP* | *LP-PL* | *PL* | **P** | **N** | **P × N** |
| 1st harvest, g | 0.6 $^{aA}$ (0.02) | 0.4 $^{abA}$ (0.02) | 0.4 $^{bA}$ (0.04) | 1.1 $^{aB}$ (0.2) | 1.5 $^{aB}$ (0.1) | 0.4 $^{bA}$ (0.1) | 1.4 $^{aB}$ (0.1) | 1.8 $^{aB}$ (0.1) | 0.6 $^{bA}$ (0.1) | *** | *** | *** |
| 2nd harvest, g | 0.2 $^{A}$ (0.03) | 0.3 $^{A}$ (0.1) | 0.3 $^{A}$ (0.1) | 1.8 $^{B}$ (0.3) | 1.8 $^{B}$ (0.2) | 1.5 $^{B}$ (0.4) | 2.6 $^{B}$ (0.1) | 2.7 $^{B}$ (0.9) | 1.8 $^{AB}$ (0.2) | ns | *** | ns |
| 3rd harvest, g | 0.6 $^{A}$ (0.1) | 0.5 $^{A}$ (0.2) | 0.5 $^{A}$ (0.04) | 4.5 $^{aB}$ (0.3) | 4.0 $^{abB}$ (0.4) | 2.6 $^{bB}$ (0.2) | 4.6 $^{B}$ (0.6) | 5.0 $^{B}$ (0.1) | 4.3 $^{C}$ (0.3) | ns | *** | * |
| Total harvest, g | 1.3 $^{A}$ (0.1) | 1.2 $^{A}$ (0.2) | 1.2 $^{A}$ (0.1) | 7.4 $^{B}$ (0.7) | 7.3 $^{B}$ (0.6) | 4.5 $^{B}$ (0.6) | 8.6 $^{B}$ (0.6) | 9.5 $^{B}$ (0.9) | 6.6 $^{B}$ (0.3) | ns | *** | ** |

*LP* represents *Lolium perenne; PL* represents *Plantago lanceolata.* SEM = standard error of the mean (in parenthesis); means with different letters are different at $p < 0.05$, $^{ab}$ represent the effect of sward type and $^{ABC}$ represent the effect of fertilization rate. $^{ns}$ $p > 0.05$, * $p < 0.05$, ** $p < 0.01$ and *** $p < 0.001$.

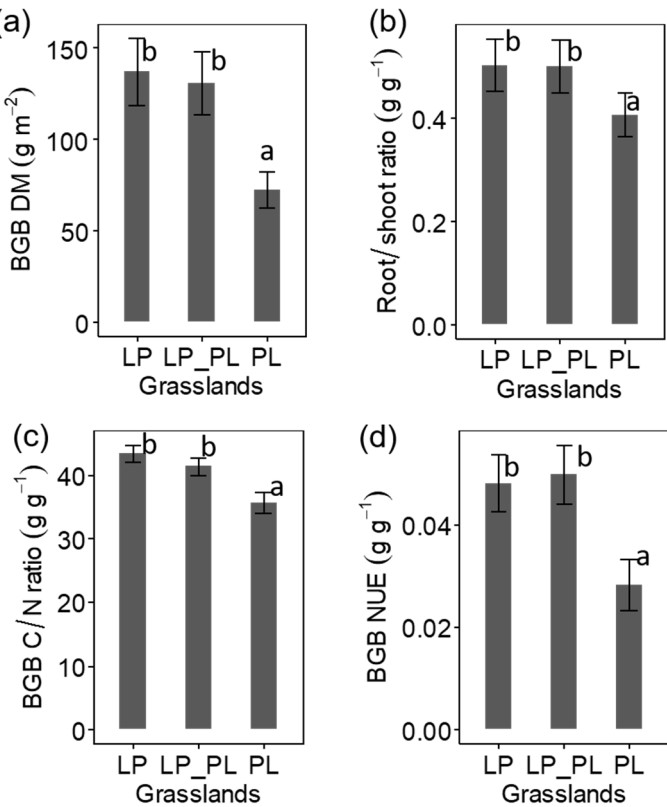

**Figure A2.** Sward effect on (**a**) the belowground (BGB) DM yield, (**b**) root/shoot ratio, (**c**) BGB C/N ratio and (**d**) BGB nitrogen use efficiency (NUE). Differing letters indicate significant differences ($p < 0.05$); error bars indicate standard errors of the mean.

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
