# Peer review of "Nitrous Oxide Emission from Forage Plantain and Perennial Ryegrass Swards Is Affected by Belowground Resource Allocation Dynamics"

_agronomy, doi:10.3390/agronomy11101936_

Round 1
Reviewer 1 Report
Dear authors,
After carefully read your work, these are my proposal to improve the quality of your article, and some questions that, in my opinion, should be clarify.
The topic of the work is interesting for me and, probably, will be also interesting for the readers of Agronomy. Althought, in my opinion, it could have been more complete by including afordable and easy measurements such as soil ammonium and nitrate content.
This reviewer should indicate the lack of enough knowledge about enzymatic analysis, so cannot enter into an in-depth assessment of the applied methodology and the interpretation of the results about these questions. This reviewer expertise are soil N2O emissions and will therefore focus on this issue.
In my opinión, the introduction lacks of some comments about the main processes determining the nitrogen fate in the soil: nitrification and denitrification.
- Line 73: the authors make reference to N2O-consuming enzymes, term which, in my opinion, is not correct here since there is only one enzyme able to consume N2O by reducing it to N2 (nitrous oxide reductase; N2OR). This enzyme has not been described in the introduction nor analyzed in this work.
- Line 77-79: I’m not agree with this statement: a larger microbial population encompasses a greater nitrifying and denitrifying activity. In addition, it is known that, as greater is de nitrate content in the soil, lower is the reduction of N2O to N2 by N2OR (Blackmer and Bremmer, 1978; https://doi.org/10.1016/0038-0717(78)90095-0; Saggar et al., 2013; https://doi.org/10.1016/j. scitotenv.2012.11.050). This leads into an incomplete denitrification, which would result in greater N2O emissions and not lower.
- Line 83-85: In my opinion, these lines are much more relevant to this study than some other parts of the introduction, so the introduction regarding the effect of mixing different species on N2O emissions should be more detailed.
- Line 103: I suppose that the soil should be dried before sieving. Is this correct? In this case, for how long was the soil storage before the start of the experiment? Did you activate the soil before the application of fertilizers?
- Line 111: this is not clear for me. Were there some pots with 32 kg ha of L. Perenne and other pots with 32 kg ha L. Perenne + 1 plant of P. Lanceolata? If this is the case, I think it is a mistake since in the mixture pots there is one more plant taking nitrogen in comparison with the pots containing only L. Perenne. So the comparison would be misleading.
- Please, include in the materials & methods section the time elapsed between each fertilizer application.
- Line 122: when analyzing N2O emissions from soil, it is very informative to know the soil water content in terms of Water Filled Pore Spaces (WFPS), since it is very informative of the state of oxygenation of the soil and, then, allow the interpretation of the origin of the emissions (nitrification or denitrification). Could you please provide this information?
- Line 127: how many days after fertilizer application was the first gaseous sample collected? Did you collect gaseous samples every day? Was the sampling carried out always and the same time?
- Line 266: Please, indicate how many days after fertilizer application did the maximum emission peak take place. Was there a time difference in the maximum peak between the different plantations?
- Line 325: I do not understand this statement considering that the authors did not measure any other nutrient uptake apart from nitrogen. Moreover, there is no difference in roots biomass between the treatments receiving 150 and 300 kg N ha.
- Line 399: In which conditions would you expect the exudation of BNIs? It is usually found under stress conditions or with low nitrogen availability, which was not the case of this study.
- Line 330-333: it seems there is a contradiction between this statement (“the presence of the plantain can increase the N use efficiency of grasslands”) and lines 404-405 (“BGB NUE was lower for P. Lanceolata swards than for L. Perenne or mixed swards”).
- Line 417: the authors cannot afirm that the effects are a consequence of the exudation of BNIs, since the presence of these compounds was not determined in this work.
- Line 470-471: the authors did not determine ammonium nor nitrate content, so this is speculative.
- Line 472-473: talking about “complete uptake” is not correct since part of the nitrogen is always lost in form of gaseous emissions.
- Line 482-483: the autors cannot include a conclusión about a factor that was not determined at all in this work. They did not determine BNI production, nor nitrifying activity nor nitrifying genes. If the objective of this work was to evaluate the nitrification inhibitory capacity of P. Lanceolata, the experimental design should have been different and with other treatments/conditions.
Author Response
Dear reviewer, please check the attachment.

Reviewer 2 Report
This manuscript is very well written and presented. The topic is interesting and valid. There are some minor changes throughout the manuscript as well as some formatting issues that should be addressed. Please refer to the track changes comment as well as the Author Guidelines for Publication.

Author Response

(The authors gave the same response as above.)

Reviewer 3 Report
Suggestions:
- improve the English language
- include 'R' script
- improve visualization of graphs
- use abbreviation of plant species
- major suggestions mentioned in the attachment

Author Response

(The authors gave the same response as above.)
